# Enhanced Antitumor Efficacy of PhAc-ALGP-Dox, an Enzyme-Activated Doxorubicin Prodrug, in a Panel of THOP1-Expressing Patient-Derived Xenografts of Soft Tissue Sarcoma

**DOI:** 10.3390/biomedicines10040862

**Published:** 2022-04-06

**Authors:** Britt Van Renterghem, Agnieszka Wozniak, Ludovica Tarantola, Andrea Casazza, Jasmien Wellens, Madita Nysen, Ulla Vanleeuw, Che-Jui Lee, Geert Reyns, Raf Sciot, Nele Kindt, Patrick Schöffski

**Affiliations:** 1Laboratory of Experimental Oncology, Catholic University of Leuven, 3000 Leuven, Belgium; britt.vanrenterghem@kuleuven.be (B.V.R.); agnieszka.wozniak@kuleuven.be (A.W.); ludovica.tarantola@gmail.com (L.T.); wellens.jasmien88@gmail.com (J.W.); madita.nysen@kuleuven.be (M.N.); ulla.vanleeuw@kuleuven.be (U.V.); jerry.lee@kuleuven.be (C.-J.L.); 2CoBioRes, Campus Gasthuisberg, Catholic University of Leuven, 3000 Leuven, Belgium; andreacasazza75@gmail.com (A.C.); reynsg@hotmail.com (G.R.); nele.kindt@cobiores.be (N.K.); 3Department of Pathology, University Hospitals Leuven, 3000 Leuven, Belgium; raf.sciot@kuleuven.be; 4Department of General Medical Oncology, University Hospitals Leuven, 3000 Leuven, Belgium

**Keywords:** PhAc-ALGP-Dox, tetra-drug technology, doxorubicin prodrug, soft tissue sarcoma, patient-derived xenograft

## Abstract

Despite poor response rates and dose-limiting cardiotoxicity, doxorubicin (doxo) remains the standard-of-care for patients with advanced soft tissue sarcoma. We evaluated the efficacy of two tetrapeptidic doxo prodrugs (PhAc-ALGP-Dox or CBR-049 and CBR-050) that are locally activated by enzymes expressed in the tumor environment, in ten sarcoma patient-derived xenografts. Xenograft models were selected based on expression of the main activating enzyme, i.e., thimet oligopeptidase (THOP1). Mice were either randomized to vehicle, doxo, CBR-049 and CBR-050 or control, doxo, aldoxorubicin (aldoxo) and CBR-049. Treatment efficacy was assessed by tumor volume measurement and histological assessment of ex-mouse tumors. CBR-049 showed significant tumor growth delay compared to control in all xenografts investigated and was superior compared to doxo in all but one. At the same time, CBR-049 showed comparable efficacy to aldoxo but the latter was found to have a complex safety profile in mice. CBR-050 demonstrated tumor growth delay compared to control in one xenograft but was not superior to doxo. For both experimental prodrugs, strong immunostaining for THOP1 was found to predict better antitumor efficacy. The prodrugs were well tolerated without any adverse events, even though molar doses were 17-fold higher than those administered and tolerated for doxo.

## 1. Introduction

Soft tissue sarcomas (STS) represent a heterogeneous group of rare, malignant tumors of mesenchymal origin, comprising more than 70 histological subtypes but accounting for only 1% of all adult malignancies [1,2]. Nevertheless, this rare family of diseases presents a major clinical challenge because of its aggressive behavior and poor prognosis, especially in case of unresectable or metastatic disease. The first-line therapy for patients in the advanced setting is doxorubicin (doxo), which has been the standard-of-care already for more than five decades [3]. Objective responses to doxo are seen in only 14–18% of treated patients according to recent clinical trials [4,5,6,7]. Meanwhile, doxo therapy is associated with dose-limiting and potentially irreversible cardiotoxicity, not allowing to exceed 550 mg/m^2^ of cumulative dose over the lifetime of a patient [8]. As a consequence, therapy is often discontinued even in the few responding patients and cannot be resumed in case of relapse.

Several efforts have been made in the past to increase tumor specificity and overcome the cardiotoxicity of doxo, such as the development of the structurally related compounds epirubicin and mitoxantrone or pegylated liposomal doxo formulations, but so far none of these agents have been able to overcome its limitations while maintaining efficacy [9,10,11]. Aldoxorubicin (aldoxo) is a prodrug of doxo that binds serum albumin through an integrated acid-sensitive linker, thereby increasing its half-life time and tumor specificity as the linker is mainly hydrolyzed in the acidic environment of the tumor. Early studies with aldoxo in rodents demonstrated a promising reduction of cardiotoxicity compared to conventional doxo [12]. In patients, aldoxo was well tolerated up to doses of 450 mg/m² (corresponding to 3.5-fold the standard dose of doxo) but failed to show superior progression-free survival in patients with advanced STS, with exception of the subgroup of patients with leiomyosarcoma and liposarcoma (L-sarcomas) [13,14].

Another approach is the development of doxo prodrugs that are predominantly activated in the vicinity of the tumor. Phosphonoacetyl-L-alanyl-L-leucyl-L-glycyl-L-prolyl-doxorubicin (PhAc-ALGP-Dox or CBR-049) and CBR-050 (proprietary peptide sequence confidential) are two such prodrugs of doxo developed by CoBioRes, Leuven, Belgium. Since their tetrapeptide sequence prevents cellular uptake, the prodrugs are biologically inactive in their uncleaved form. Only when cleaved by extracellular thimet oligopeptidase (THOP1), the inactive intermediate GP-doxo diffuses into the cell where it is cleaved by cytoplasmic fibroblast activation protein (FAP) and/or dipeptidyl peptidase-4 (DPP4) to release active doxo [15]. Considering that the expression of THOP1 has been repeatedly found upregulated in cancer cells compared to normal cells, including sarcomas [15,16,17,18,19], the prodrugs have potentially increased tolerability, tumor specificity and efficacy compared to conventional doxo. Cornillie et al. previously documented superior efficacy of CBR-049 compared to doxo, as delivered by intraperitoneal minipump in three sarcoma patient-derived xenograft (PDX) models [19]. Since the clinical applicability of this route of administration was questionable and information on THOP1-related sensitivity was lacking, we further investigated intravenous administration of both CBR-049 and CBR-050 (another CoBioRes lead molecule) in 10 THOP1-expressing sarcoma xenograft models.

## 2. Materials and Methods

### 2.1. Patient-Derived Sarcoma Xenograft Models

Xenograft models used in this study were selected from the extensive “XenoSarc” platform of PDX models, available in the Laboratory of Experimental Oncology, Catholic University Leuven (KU Leuven), Leuven, Belgium. Models were established by subcutaneous transplantation of fresh human tumor tissue in partially immunodeficient NMRI *nu/nu* mice (Janvier Labs, Le Genest-Saint-Isle, France) as previously described [20]. Donor tissue was obtained from patients with STS who underwent a routine biopsy or surgery at the University Hospitals Leuven (UZ Leuven), Leuven, Belgium. Written informed consent was obtained from each patient for xenografting of leftover tissue to create PDX models and subsequent use of these models for translational research purposes. All procedures were approved by the Medical Ethics Committee, UZ Leuven (project S53483) and the Ethics Committee for Laboratory Animals, KU Leuven (project P175-2015).

For the current project, xenografts were selected based on their level of THOP1 expression as determined by enzyme-linked immunosorbent assay (ELISA) on archived frozen tumor fragments, using the RayBio^®^ Human THOP1 enzyme-kit (Raybiotech, Georgia, GA, USA) according to manufacturer’s protocol. Total protein extracts were obtained with radioimmunoprecipitation assay (RIPA) buffer, supplemented with Pierce protease and phosphatase inhibitor cocktail (ThermoFisher Scientific, Waltham, MA, USA). For each sample, 40 µg of total protein extract was loaded in duplicate on pre-coated plates for quantification of the human THOP1 expression.

Six xenograft models of synovial sarcoma (UZLX-STS7^SynSa^), leiomyosarcoma (UZLX-STS22^LMS^ and -STS128^LMS^), undifferentiated pleomorphic sarcoma (UZLX-STS84^UPS^), myxofibrosarcoma (UZLX-STS89^MFS^) and intimal sarcoma (UZLX-STS185^IS^) were chosen for comparison of CBR-049 and CBR-050 against doxo. Four additional xenografts of dedifferentiated liposarcoma (UZLX-STS204^DDLPS^), undifferentiated pleomorphic sarcoma (UZLX-STS211A^UPS^), myxofibrosarcoma (UZLX-STS216^MFS^) and extraskeletal osteosarcoma (UZLX-STS234^eOS^) were selected for comparison of CBR-049 against doxo and aldoxo. All xenografts demonstrated THOP1 expression at different levels ranging from 23 to 299 ng/mg total protein (Appendix A).

Xenograft models were characterized using histological and molecular methods as previously described [20]. Briefly, tumor tissue from each passage was characterized morphologically and molecularly by hematoxylin and eosin (H&E) staining, subtype-specific immunohistochemistry (IHC) and/or fluorescence in situ hybridization (FISH), under supervision of a sarcoma reference pathologist (R.S.). Only after observing stable characteristics for at least two passages, a model was considered established and used for experiments. Of note, models UZLX-STS7^SynSa^, -STS22^LMS^ and -STS84^UPS^ have been used previously for in vivo drug testing [19,21].

### 2.2. Drugs and Reagents

All drugs were provided by CoBioRes. CBR-049, CBR-050 and doxo were supplied as working solution in sterile 0.9% saline. Aldoxo was provided as powder and reconstituted in glucose phosphate buffer (10 mM sodium phosphate, 5% glucose, pH6.4) right before injection.

The following antibodies were used for IHC: alpha-smooth muscle actin (alpha-SMA; #M085129-2, Agilent Technologies, Santa Clara, CA, USA), cleaved poly (ADP-ribose) polymerase (cleaved PARP; #Ab32064, Abcam, Cambridge, UK), desmin (#SP138, ThermoFisher Scientific), murine double minute 2 homolog (MDM2; #337100, ThermoFisher Scientific), phospho-histone H3 (pHH3; #9701L, Cell Signaling Technology, Danvers, MA, USA), THOP1 (ab154173, Abcam) and transducin-like enhancer of split 1 (TLE-1, #ab183742, Abcam). All sections were incubated with secondary antibody-horseradish peroxidase polymer conjugate (Envision+ System-HRP, Agilent Technologies), except for cleaved PARP where SignalStain Boost IHC Detection Reagent (Cell Signaling Technology) was used. Stainings were developed using diaminobenzidine (DAB), followed by hematoxylin counterstaining (VWR, Radnor, PA, USA).

The following molecular probes were used for FISH: Vysis SS18 Break Apart FISH Probe Kit (Abbott, Lake Bluff, IL, USA) and Kreatech™ FISH probes MDM2 (12q15)/SE 12 (D12Z3) (Leica Biosystems, Chicago, IL, USA). Tissue pretreatment, hybridization and detection were carried out according to the manufacturer’s instructions.

### 2.3. Experimental Setup

Xenograft models were expanded for the creation of an experimental cohort by unilateral subcutaneous transplantation of tumor tissue from the previous passage to the left flank of NMRI nu/nu mice. After approximately two to three weeks, mice with growing tumors were randomly divided into the four different treatment groups.

#### 2.3.1. CBR-049 and CBR-050 vs. Doxo

Mice received either 0.9% saline (vehicle, control), doxo (8.6 µmol/kg), CBR-049 (150 µmol/kg) or CBR-050 (150 µmol/kg) and were sacrificed two weeks after their last treatment (day 36) for histological assessment of the ex-mouse tumors.

#### 2.3.2. CBR-049 vs. Doxo and Aldoxo

Mice received either glucose phosphate buffer (vehicle, control), doxo (8.6 µmol/kg), aldoxo (32 µmol/kg) or CBR-049 (150 µmol/kg). All mice of the control group and part of mice from the actively treated groups were sacrificed one week after their last treatment (day 29, short-term experiment) for histological assessment. Remaining mice were kept until day 85 (long-term experiment) to follow post-treatment tumor evolution.

All treatments were administered weekly for four weeks (day 1, 8, 15 and 22) by intravenous tail vein injection. Relative tumor volume and body weight of every animal were monitored three times per week by normalizing the value on a given day (day x) against the value at baseline (day 1), i.e., day x/day 1 × 100. Tumor was measured three dimensionally by caliper and tumor volume was calculated using the following formula: tumor volume (mm^3^) = length (mm) × width (mm) × height (mm). During the experiment, mice with tumor volume > 2000 mm^3^, body weight loss > 20% or other severe adverse events were sacrificed because of reaching the humane endpoint. Detailed description on the number of mice/tumors included in each experiment are shown in Appendix A.

Tumor growth delay compared to control was compared to the respective THOP1 expression level of each xenograft as determined by ELISA (performed on archived tissue for model selection) and IHC (performed on experimental control tumors). THOP1 immunostainings were given a number from 0 to 3 as determined by overall intensity of the staining (0: negative; 1: weakly positive; 2: intermediate positive; 3: strongly positive). Tumor growth delay was calculated (day 29 for all experiments) using the following formula: ((C-T/C) × 100), where C and T are the average relative tumor volumes of the control and treated groups, respectively.

### 2.4. Histological Assessment

Tumors collected for histological assessment were cut into 4-µm sections after formalin-fixation and paraffin embedding. Sections were microscopically evaluated to confirm model characteristics (control tumors) and to evaluate the treatment efficacy in terms of proliferation and apoptosis. Mitotic and apoptotic activity were assessed on H&E-stained sections by counting mitotic and apoptotic figures in 10 high power fields (HPF) at 400-fold magnification (0.45-mm field diameter). pHH3 and cleaved PARP stainings, as markers for proliferation and apoptosis, were assessed by counting the number of positive tumor cells in 10 HPF. Histological analysis was performed blindly for treatment groups using a CH30 microscope (Olympus, Tokio, Japan). Pictures were taken using a BX43 light microscope (Olympus).

### 2.5. Statistics

Statistical analysis was performed using the GraphPad Prism software (version 8.4.3, San Diego, CA, USA). The Mann–Whitney U-test (MWU) was applied for comparison of tumor volume and histological assessment between the different treatment groups. *p* < 0.05 was considered as statistically significant. Data from dropout animals were excluded from the statistical analysis but included in the graphs for tumor volume and body weight.

## 3. Results

### 3.1. Characterization of the Selected STS Xenograft Models

Original patient tumors and corresponding experimental ex-mouse tumors shared the same morphological and molecular features (Figure 1). UZLX-STS7^SynSa^ showed bundles of spindle cells and synovial sarcoma-specific TLE-1 immunopositivity and SS18 rearrangement. Both UZLX-STS22^LMS^ and -STS128^LMS^ demonstrated spindle to epithelioid cells positive for the smooth-muscle cell marker alpha-SMA, but only UZLX-STS22^LMS^ showed positivity for desmin, which corresponds with their original donor samples. UZLX-STS84^UPS^ and UZLX-STS211A^UPS^ showed atypical pleomorphic cells with diffuse areas of necrosis. UZLX-STS89^MFS^ and UZLX-STS216^MFS^ showed spindle to epithelioid shaped cells in a myxoid background. Both UZLX-STS185^IS^ and -STS204^DDLPS^ demonstrated spindle to epithelioid cells with nuclear expression of MDM2. While FISH analysis of UZLX-STS204^DDLPS^ demonstrated MDM2 amplification characteristic for dedifferentiated liposarcoma, analysis in UZLX-STS185^IS^ showed MDM2 polysomy as observed in the tumor obtained from the donor patient. UZLX-STS234^eOS^ showed spindle-shaped cells with diffuse areas of osteoid production and calcification. All models showed immunopositivity for THOP1, except for UZLX-STS84^UPS^ showing also the lowest THOP1 expression by ELISA (Appendix A).

### 3.2. CBR-049 and CBR-050 vs. Doxo

#### 3.2.1. Tumor Volume Evaluation

At the end of the experiments (day 36), control tumors from all six models demonstrated a steady increase in relative tumor volume, reaching 1801% of baseline tumor volume in UZLX-STS7^SynSa^, 1098% in -STS22^LMS^, 2800% in -STS84^UPS^, 898% in -STS89^MFS^, 775% in -STS128^LMS^ and 2804% in -STS185^IS^ (Figure 2 and Appendix A). Doxo-treated tumors showed a steady increase in tumor volume in UZLX-STS22^LMS^, -STS84^UPS^ and -STS128^LMS^ (962%, 1579% and 508% of baseline tumor volume) but tumor growth was significantly delayed compared to control in UZLX-STS7^SynSa^, -STS89^MFS^ and -STS185^IS^ (135%, 154% and 938% of baseline tumor volume). CBR-049-treated tumors showed significantly delayed tumor growth compared to control in all six xenograft models and compared to doxo in UZLX-STS22^LMS^, -STS84^UPS^, -STS89^MFS^, -STS128^LMS^ and -STS185^IS^, with 187% of baseline tumor volume in UZLX-STS7^SynSa^, 190% in -STS22^LMS^, 486% in -STS84^UPS^, 51% in -STS89^MFS^, 62% in -STS128^LMS^ and 444% in -STS185^IS^. In contrast, CBR-050-treated tumors demonstrated significantly delayed tumor growth only in UZLX-STS89^MFS^ compared to control but not to doxo with 238% of baseline tumor volume. Individual tumor growth curves for each experiment are shown in Appendix A. All treatments were well tolerated with mice showing stable body weight within ethically acceptable limits (Appendix A). Besides a red but harmless urine discoloration 1–2 days post administration, no other side-effects were observed upon doxorubicin or prodrugs therapy in mice. Macroscopic evaluation of the heart, liver and kidneys upon sacrification did not reveal any alteration nor signs of toxicity.

#### 3.2.2. Histological Assessment

On the day of tumor collection (day 36), control tumors from all six models demonstrated high mitotic activity with an average of more than 30 mitotic figures per 10 HPF on both H&E and pHH3 staining (Figure 3A). On H&E, doxo-treated tumors showed a significantly reduced number of mitotic cells compared to control in UZLX-STS7^SynSa^ and -STS89^UPS^, which was further confirmed by pHH3 staining. CBR-049-treated tumors showed significantly reduced mitotic count compared to control in all evaluable models and compared to doxo in all but one (UZLX-STS89^UPS^). Of note, tumors from UZLX-STS7^SynSa^ could not be assessed because of a technical problem that occurred during the embedding of the formalin-fixed tumors in paraffin. In contrast to CBR-049, CBR-050-treated tumors demonstrated a significantly reduced number of mitotic cells compared to control only in UZLX-STS89^MFS^, though this was not significant in comparison to doxo. These results are fully in line with the findings of tumor growth evaluation. Apoptotic activity was not found altered in any of the treatment groups on the day of tumor collection, two weeks after the last treatment (Figure 3B). In order to increase the chance of capturing a change in apoptotic activity for further experiments, parts of tumors were collected one week earlier (day 29) and remaining mice were kept until day 85 to follow post-treatment tumor evolution.

### 3.3. CBR-049 vs. Doxo and Aldoxo

#### 3.3.1. Tumor Volume Evaluation

At the end of the short-term experiments (day 29), control tumors from all four models demonstrated a steady increase in relative tumor volume, reaching 559% of baseline tumor volume in UZLX-STS204^DDLPS^, 657% in -STS211A^UPS^, 786% in -STS216^MFS^ and 453% in -STS234^eOS^ (Figure 4A and Appendix A). Doxo-treated tumors showed a steady increase in tumor volume in UZLX-STS204^DDLPS^, -STS216^MFS^ and -STS234^eOS^ (276%, 491% and 451% of baseline tumor volume) but tumor growth was significantly delayed compared to control in UZLX-STS211A^UPS^ (502% of baseline tumor volume). Aldoxo-treated tumors showed significantly delayed tumor growth compared to control and doxo in all four models with 125% of baseline tumor volume in UZLX-STS204^DDLPS^, 275% in UZLX-STS211A^UPS^, 82% in UZLX-STS216^MFS^ and 257% in UZLX-STS234^eOS^. Similarly, CBR-049-treated tumors demonstrated significantly delayed tumor growth compared to control and doxo in all four models with 123% of baseline tumor volume in UZLX-STS204^DDLPS^, 265% in UZLX-STS211A^UPS^, 176% in UZLX-STS216^MFS^ and 238% in UZLX-STS234^eOS^.

At the end of the long-term experiments (day 85), doxo-treated tumors showed tumor volume increase even after initial delay (Figure 4B). At the same time, both aldoxo- and CBR-049-treated tumors maintained comparable tumor volume delay until day 85. In one model (UZLX-STS216^MFS^) CBR-049 demonstrated a slightly more pronounced effect over aldoxo. Again, all treatments were well tolerated except for aldoxo where several animals developed hind limb paralysis causing problems with weight maintenance, which lead to drop out of multiple animals (Appendix A).

#### 3.3.2. Histological Assessment

On the day of tumor collection (day 29), control tumors of all models demonstrated high mitotic activity with an average of more than 10 mitotic figures per 10 HPF on H&E and pHH3 staining (Figure 5A). Doxo-treated tumors did not show a reduced number of mitotic cells compared to control in any of the models. On H&E, aldoxo-treated tumors showed a significantly reduced number of mitotic cells compared to control and doxo in UZLX-STS204^DDLPS^ and -STS211A^UPS^, which was also confirmed by pHH3 staining. Tumors from UZLX-STS216^MFS^ could not be assessed since they were too small to count 10 HPF. CBR-049-treated tumors demonstrated significantly decreased mitotic count compared to control and doxo in all four models investigated and compared to aldoxo in UZLX-STS234^eOS^. As tumors were now collected one week earlier, increased apoptotic count could be demonstrated in aldoxo-treated tumors of UZLX-STS204^DDLPS^ and -STS211A^UPS^ compared to control and doxo (Figure 5B). CBR-049-treated tumors showed significantly increased apoptosis compared to control in all four models on cleaved PARP immunostaining and compared to doxo and aldoxo in UZLX-STS234^eOS^. Again, these results reflect the tumor growth evolution seen in these models.

### 3.4. THOP1 Expression and Tumor Growth Delay

Tumor growth delay compared to control at day 29 of the experiments ranged from 47% to 95% for the CBR-049-treated xenografts and 11% to 87% for the CBR-050-treated xenografts (Appendix A). Although xenografts showed different THOP1 expression levels ranging from 23 to 299 ng/mg total protein as determined by ELISA, higher protein levels did not per se result in more pronounced tumor growth delay and vice versa. Meanwhile, a trend towards more pronounced tumor growth delay could be observed for models with intermediate to strong THOP1 immunostaining. Best results were seen for most strongly positive models UZLX-STS89^MFS^ and -STS185^IS^ with tumor growth delay of, respectively, 94% and 86% upon CBR-049 and 87% and 60% upon CBR-050. The only negative xenograft (UZLX-STS84^UPS^) was also lowest on ELISA and showed limited tumor growth delay of 53% and 30% for CBR-049 and CBR-050, respectively.

## 4. Discussion

Despite poor response rates and dose-limiting cardiotoxicity, doxo remains the standard-of-care for patients with advanced sarcoma for already more than five decades [4]. Although a high number of promising agents were tested over the years, only trabectedin, eribulin and pazopanib achieved regulatory approval for treatment of patients with advanced STS as second-line therapy after failure of doxo with objective response rates in the single digit range [22,23,24]. Hence, the need for alternative more active and—ideally—less toxic treatment options for these patients remains unfulfilled.

We investigated the efficacy of two innovative tetrapeptidic doxo prodrugs (CBR-049 and CBR-050) that are locally activated by enzymes expressed in the tumor environment, in sarcoma patient-derived xenografts. Since the prodrugs mostly remain inactive in the blood and normal tissues, they have improved tolerability as demonstrated by the increased median lethal dose (LD_50_) and maximum tolerated dose (MTD) and decreased hematologic and cardiac toxicity of CBR-049 compared to doxo [15]. Consequently, higher doxo-equivalent concentrations can be administered in vivo, which together with their favorable tissue distribution, will result in higher local concentrations of active doxo at the tumor site. Whether this also translates into increased antitumor efficacy of the prodrugs was demonstrated by Cornillie and colleagues, who evaluated CBR-049 delivered by intraperitoneal minipump at 30- to 40-fold higher molar doses than those used for doxo (cumulative dose over 7 days of 1.2 mmol/kg) in two xenografts of dedifferentiated liposarcoma (UZLX-STS3^DDLPS^ and UZLX-STS5^DDLPS^) and one synovial sarcoma (UZLX-STS7^SynSa^), whereof the latter xenograft is also included in the current study. In this initial pilot study, CBR-049 revealed superior tumor growth delay compared to doxo (cumulative dose over 7 days of 0.03 mmol/kg in UZLX-STS3 and 0.04 mmol/kg in UZLX-STS5 and -STS7) in all three xenografts [19]. Since the clinical applicability of this route of administration was questionable and information on THOP1-related sensitivity was still lacking, we further investigated intravenous administration of both CBR-049 and CBR-050 (another lead molecule from CoBioRes) in sarcoma xenografts, all selected based on expression of the main activating enzyme THOP1.

In a first step, we compared the efficacy of both prodrugs against vehicle (control) and doxo in six xenografts at molar doses more than 17-fold higher than those tolerated and administered for doxo. CBR-049 showed delayed tumor growth compared to control in 6/6 xenografts investigated and in 5/6 compared to doxo, including 2/3 doxo-responsive xenografts (UZLX-STS89^MFS^ and -STS185^IS^ but not UZLX-STS7^SynSa^). CBR-050 delayed tumor growth significantly in 1/6 xenografts but only when compared to control and not to doxo. Of note, the results obtained with doxo in UZLX-STS7^SynSa^ in the current study are in contrast with the results obtained by Cornillie and colleagues in the same model, where it did not respond to doxo by intraperitoneal minipump administration and did show superior efficacy of CBR-049 compared to doxo [19]. In the current study, however, CBR-049 failed to show superiority compared to doxo due to extreme sensitivity of this model to doxo alone. The conflicting results observed in this model are likely related to differences in doxo absorption and stability during minipump administration, since the cumulative dose of doxo was approximately the same in both experiments (0.034 mmol/kg in this study). While intraperitoneal drug administration is common practice in rodent studies because of its convenience and suitability for a variety of formulations (e.g., amenable to physicochemical properties of a drug), it should only be considered for proof-of-concept studies with non-irritable compounds due to its main problems with bioavailability (high first pass metabolism), risk of (fatal) chemical peritonitis and limited clinical applicability [25]. The choice for intravenous administration of doxo in the current study was taken upon consideration of its severe, necrotizing local toxicity and application in the clinic [26].

Since CBR-049 clearly outperformed its analogue CBR-050 in the first part of this study, we decided to perform additional experiments with only CBR-049 and compare its activity with aldoxo, a competitor prodrug currently under clinical investigation for the treatment of STS [14]. For these experiments, another four STS xenografts were selected based on their THOP1 expression. Again, 4/4 xenografts showed tumor growth delay upon CBR-049 treatment compared to control and doxo, including the only doxo-responsive xenograft (UZLX-STS211A^UPS^). In the meantime, CBR-049 demonstrated comparable activity as aldoxo but several mice treated with aldoxo showed hind limb paralysis causing problems with weight maintenance, which even lead to sacrification in a few severe cases. Similar observations were made in CD1-mice treated with a single dose of aldoxo (>30 mg/kg) where motoric disturbances and peripheral neuropathy were observed in the majority of mice starting 1–3 weeks after intravenous injection [27]. Both adverse events have been well documented in mice following treatment with doxo, though not observed in doxo-treated mice in this study with a recommended dose of 8.6 mmol/kg or 5 mg/kg/week. Another study in nude mice applying the same dosage for aldoxo as used in this study (32 mmol/kg or 24 mg/kg/week) showed similar adverse events: rapid weight loss, impaired mobility, decreased food intake and signs of lethargy starting after the last intravenous administration [28]. Of note, this dose is still far below the well-tolerated dose of 350 mg/m^2^ administered in patients from the phase 2 and 3 clinical trial without manifestation of motoric or neurological events and without evidence of acute cardiotoxicity [14,29].

In a last step, we aimed to identify THOP1 as biomarker of sensitivity to the prodrugs by correlating tumor growth delay compared to control with the THOP1 expression level of each xenograft as determined by ELISA and IHC. Only the latter technique was found to predict best responses to the prodrugs in xenografts staining strongly positive for THOP1. Even though we would expect better correlation with a more sensitive and quantitative technique as ELISA, it should be noted that IHC was performed using an antibody recognizing both human and murine variants. Considering that the human tumor stroma is replaced by murine stroma throughout tumor growth in mice [30], part of the THOP1 expression that is produced by the tumor stroma is expected to be of murine origin. Consequently, IHC is probably giving the closest estimation of the total THOP1 level in these samples, though ELISA should still be considered for clinical samples as a more sensitive technique. To illustrate, the only immunohistochemistry negative model (UZLX-STS84^UPS^) did show a small but non-negligible amount of THOP1 upon ELISA, although an underestimation of the total THOP1 as explained above, which could be responsible for the antitumor activity seen in this xenograft. Another possibility could be the presence of a different activating enzyme than THOP1. However, the role of relevant peptidases, such as CD10, cathepsin B, MMP2 and MMP9, has been investigated in previous work, showing no relevant role in CBR-049 cytotoxicity [15].

The unique activation mode of the prodrugs has therapeutic potential besides STS due to the expression of its activating enzymes in a variety of solid tumors, including the majority of breast and prostate tumors [15,16]. The broad applicability of CBR-049 was recently demonstrated by consistent efficacy in multiple murine and human xenograft models of breast cancer, colorectal cancer, glioblastoma, non-small cell lung cancer and cisplatin-resistant ovarian cancer [15]. While its activating enzymes are also expressed to a certain extent in normal tissues, tumor specificity is obtained considering that extracellular THOP1 is almost exclusively observed in cancer cells [15]. In line with this, CBR-049 was found to have improved safety compared to conventional doxo towards normal epithelium, hematopoietic progenitors and cardiomyocytes [15]. Moreover, in this study no adverse events could be observed in mice treated with the experimental prodrugs, even though molar doses were 17-fold higher than those administered for doxo. Although the current study lacked more profound toxicological evaluation (e.g., blood analysis or histopathological evaluation of organs) and pharmacokinetic analysis, this has been covered in previous work [15].

In conclusion, CBR-049 demonstrated antitumor efficacy in all sarcoma xenografts investigated and showed superior efficacy compared to the standard-of-care doxo in all but one xenograft. CBR-049 demonstrated comparable efficacy to aldoxo, while the latter showed problematic safety profile in mice. CBR-050 demonstrated antitumor efficacy only in one sarcoma xenograft and was not superior to doxo. Strong THOP1 immunostaining was found to predict better antitumor efficacy of the prodrugs. Both experimental prodrugs were well tolerated without any adverse events, even though molar doses were 17-fold higher and a comparable dose of doxo would be lethal.

## Figures and Tables

**Figure 1 biomedicines-10-00862-f001:**
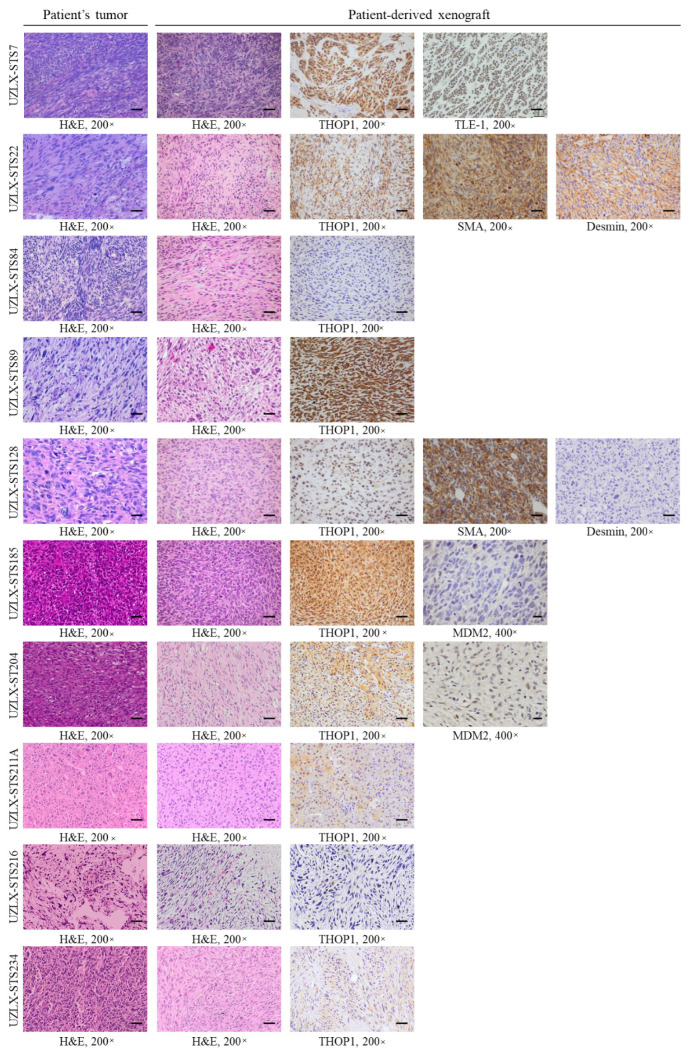
Characterization of the patient-derived sarcoma xenograft models used in this study. Representative H&E and immunostainings of the original patient tumors and the corresponding patient-derived xenografts; scale bar 200×: 50 µm, scale bar 400×: 20 µm. Alpha-SMA: alpha smooth muscle actin; H&E: hematoxylin and eosin; MDM2: mouse double minute 2 homolog; THOP1: thimet oligopeptidase; TLE-1: transducin-like enhancer of split 1; 200×: 200-fold magnification; 400×: 400-fold magnification.

**Figure 2 biomedicines-10-00862-f002:**
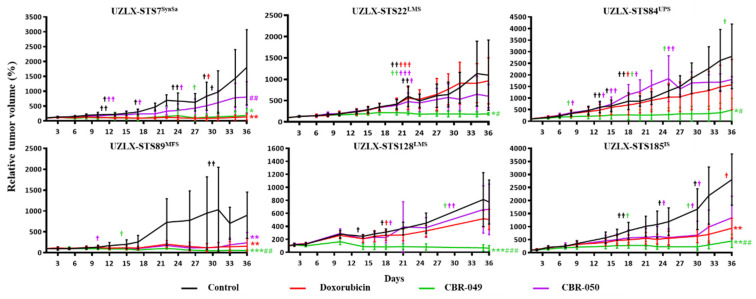
Relative tumor volume (%) compared to baseline in the patient-derived xenografts, presented as average ± standard deviation. Data from dropout animals were included in the graph but excluded from the statistical analysis. Statistical significance was calculated using Mann–Whitney U test. #†: number of mice sacrificed during the experiment per group. * *p* < 0.05 compared to control; # *p* < 0.05 compared to doxorubicin (doxo); ** *p* < 0.005 compared to control; ## *p* < 0.005 compared to doxo; *** *p* < 0.0005 compared to control; ### *p* < 0.0005 compared to doxo.

**Figure 3 biomedicines-10-00862-f003:**
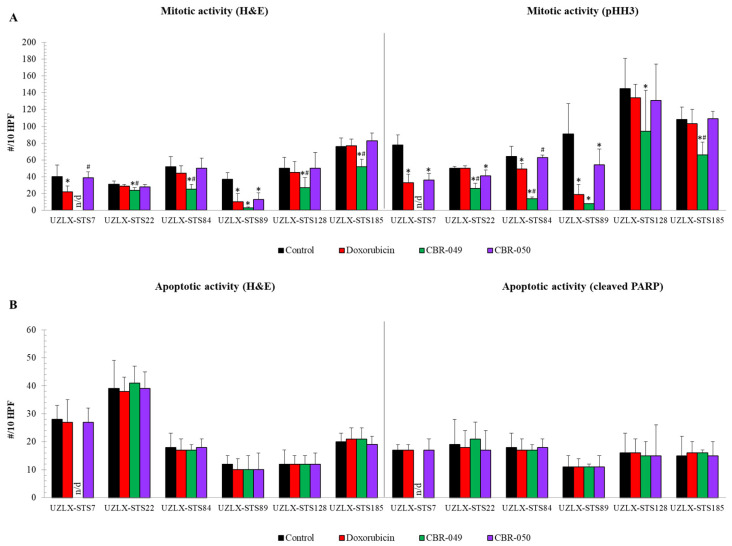
Assessment of mitotic and apoptotic activity of tumors collected at the end of experiment (day 36). (**A**) Mitotic cell count assessed on H&E and pHH3 staining; (**B**) Apoptotic cell count assessed on H&E and cleaved PARP staining. Data are presented as average ± standard deviation. CBR-049-treated samples from UZLX-STS7 could not be assessed because of a technical problem that occurred during the embedding of the formalin-fixed tumors in paraffin. * *p* < 0.05 as compared to control; # *p* < 0.05 as compared to doxorubicin. H&E: hematoxylin and eosin; pHH3: phospho-histone H3; PARP: poly (ADP-ribose) polymerase; HPF: high power fields.

**Figure 4 biomedicines-10-00862-f004:**
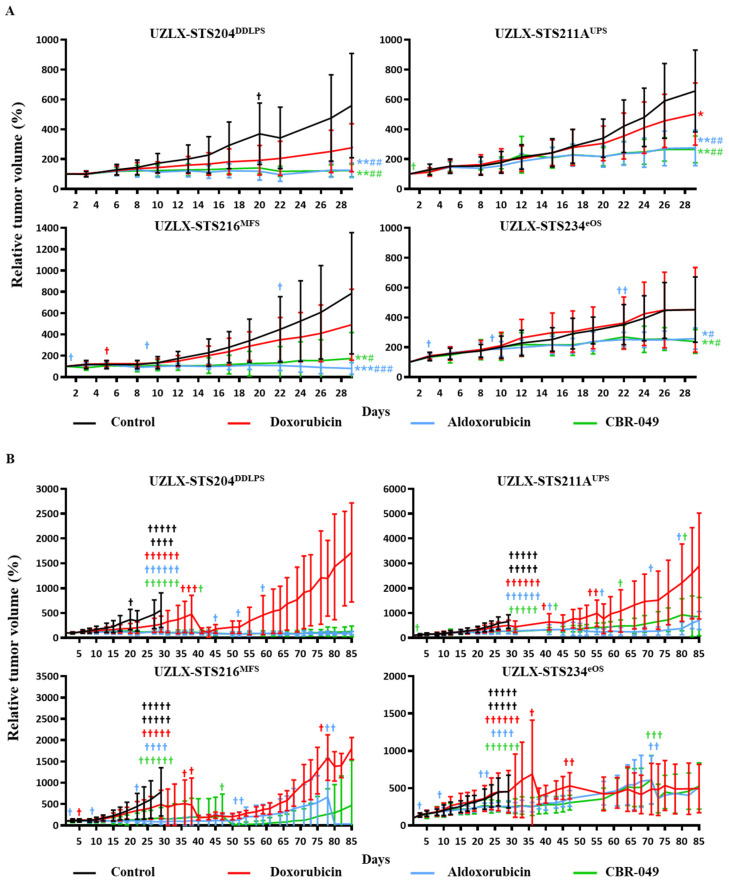
Relative tumor volume (%) compared to baseline in the patient-derived sarcoma xenografts, presented as average ± standard deviation. (**A**) Graphs until day 29 (short-term experiment); (**B**) graphs until day 85 (long-term experiment). Statistical significance was calculated using Mann–Whitney U test. #†: number of mice sacrificed during the experiment. * *p* < 0.05 compared to control; # *p* < 0.05 compared to doxorubicin (doxo); ** *p* < 0.005 compared to control; ## *p* < 0.005 compared to doxo; *** *p* < 0.0005 compared to control; ### *p* < 0.0005 compared to doxo.

**Figure 5 biomedicines-10-00862-f005:**
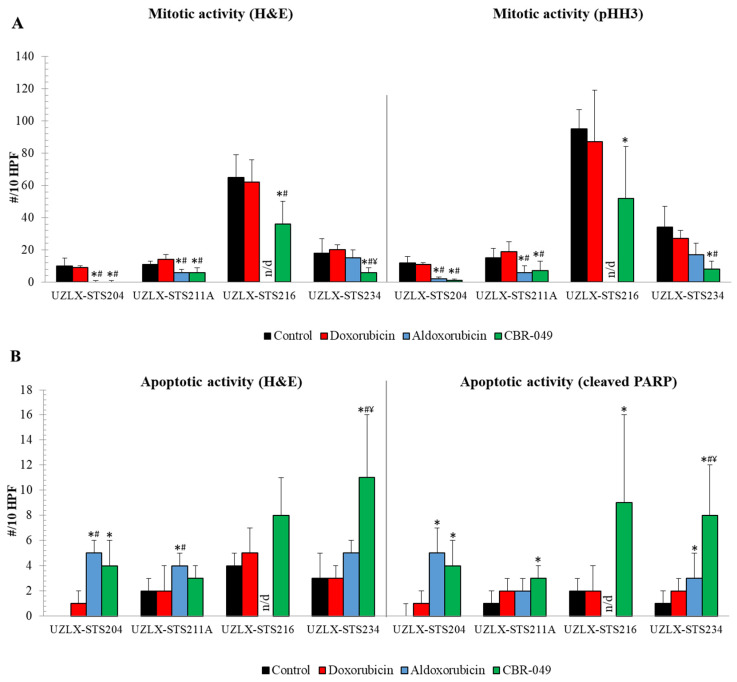
Assessment of mitotic and apoptotic activity of tumors collected on day 29 of the experiment. (**A**) Mitotic cell count assessed on H&E and pHH3 staining; (**B**) apoptotic cell count assessed on H&E and cleaved PARP staining. Data are presented as average ± standard deviation. Aldoxo-treated samples from UZLX-STS216 could not be assessed as they were too small to count 10 HPF. * *p* < 0.05 as compared to control; # *p* < 0.05 as compared to doxorubicin; ¥ *p* < 0.05 as compared to aldoxorubicin. H&E: hematoxylin and eosin; pHH3: phospho-histone H3; PARP: poly (ADP-ribose) polymerase; HPF: high power fields.

## Data Availability

Not applicable.

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
