# Peer review of "Enhanced Antitumor Efficacy of PhAc-ALGP-Dox, an Enzyme-Activated Doxorubicin Prodrug, in a Panel of THOP1-Expressing Patient-Derived Xenografts of Soft Tissue Sarcoma"

_biomedicines, 2022, doi:10.3390/biomedicines10040862_

Round 1

Reviewer 1 Report

This manuscript describes a series of experiments comparing 2 new doxorubicin prodrugs with doxorubicin and aldoxorubicin in the treatment of a series of sarcoma tumor xenografts.  The authors correctly point out that doxorubicin is still the backbone of human sarcoma chemotherapy and that there is continued need for improvement.  Their experiments support further development of their analog, CBR-049.  CBR-049 was superior to control in all cases, superior to doxorubicin in 5/6 experiments, and as effective but less toxic in their model than aldoxorubicin.  The other analog studied, CBR-050, was less effective.  CBR-049 requires cleavage at the tumor site for activation, primarily by THOP1, and they correlate the relative activity of CBR-049 in their experiments with THOP1 immunostaining.  They should discuss further the finding that CBR-049 was highly active in their one tumor that did not express THOP1.  They note that it was not superior to doxorubicin in this model, but they fail to discuss that the reason for this was not relative lack of activity of CBR-049 compared with control in this model but rather the extreme sensitivity of this model to doxorubicin alone in comparison with all of their other models.  Some discussion is required as to why a drug that requires activation would work in a tumor that cannot activate it.  Their must be a different activating enzyme in that particular tumor model.

Author Response

Indeed, CBR-049 showed good antitumor activity compared to control (and doxorubicin*) in UZLX-STS84, while THOP1 immunostaining was negative. We agree this might suggest the presence of other activating enzymes than THOP1 in this model. However, the activating potential of relevant peptidases, such as CD10, cathepsin B, MMP2 and MMP9, has been investigated in previous work recently published by Casazza and colleagues (DOI: 10.1158/1535-7163.MCT-21-0518), showing no relevant role in CBR-049 cytotoxicity. As mentioned in the manuscript, we believe it is rather a sensitivity problem of the THOP1 IHC assay, since ELISA did demonstrate a small but non-negligible amount of THOP1 that could be responsible for the limited but still significant tumor growth delay seen in this xenograft. Also mentioned in the manuscript is that ELISA is not able to detect stromal THOP1 that is of murine origin in the xenografts. Hence, we have to take into account that the amount of THOP1 detected by ELISA in the xenografts is an underestimation of the total THOP1, though still not detectable by IHC.

*Only in the THOP1-expressing xenograft UZLX-STS7 CBR-049 was not superior compared to doxorubicin due to extreme activity of doxorubicin alone in this model.

The following sentences were altered/added to the revised manuscript (discussion p.12): Even though we would expect better correlation with a more sensitive and quantitative technique as ELISA, it should be noted that IHC was performed using an antibody recognizing both human and murine variants. Considering that the human tumor stroma is replaced by murine stroma throughout tumor growth in mice [30], part of the THOP1 expression that is produced by the tumor stroma is expected to be of murine origin. Consequently, IHC is probably giving the closest estimation of the total THOP1 level in these samples, though ELISA could should still be considered for clinical samples as a more sensitive technique. To illustrate, the only immunohistochemistry negative model (UZLX-STS84UPS) did show a small but non-negligible amount of THOP1 upon ELISA, although an underestimation of the total THOP1 as explained above, which could be responsible for the antitumor activity seen in this xenograft. Another possibility could be the presence of a different activating enzyme than THOP1. However, the role of relevant peptidases, such as CD10, cathepsin B, MMP2 and MMP9, has been investigated in previous work, showing no relevant role in CBR-049 cytotoxicity [15].

The following sentence was added to the revised manuscript (discussion p. 11): Of note, the results obtained with doxo in UZLX-STS7SynSa in the current study are in con-trast with the results obtained by Cornillie and colleagues in the same model, where it did not respond to doxo by intraperitoneal minipump administration and did show superior efficacy of CBR-049 compared to doxo [19]. In the current study, however, CBR-049 failed to show superiority compared to doxo due to extreme sensitivity of this model to doxo alone. The conflicting results observed in this model are likely related to differences in doxo ab-sorption and stability during minipump administration, since the cumulative dose of doxo was approximately the same in both experiments (0.034 mmol/kg in this study).

Reviewer 2 Report

In this study Renterghem et al., reported the efficacy of two tetrapeptidic doxo prodrugs (PhAc-ALGP-Dox or CBR-049 and CBR-050) against sarcoma patient-derived xenografts. They showed CBR-049 showed significant tumor growth delay compared to doxo. Also, they reported CBR-050 demonstrated tumor growth delay compared to control in one xenograft but was not superior to doxo. Expression of THOP1 was found to predict better antitumor efficacy of both experimental prodrugs. However, authors need to address the following concerns before publication.

  1. Did the authors assess any signs of distress and dehydration effects post prodrug(s) administration in mice?

  2. What was the age of the mice at the time of PDX generation? Have authors evaluated the survival effects of prodrugs in mice beyond 28 days? If so, what probable physical changes were observed?

  3. Did authors have collected the heart, liver, spleen, lung, brain, and kidney of mice after 28 days. Did you perform any histopathology analysis, to determine the effect of prodrugs on organ functions?

  4. Did mice complete blood count analysis was performed during the course of the study? Had you looked into comprehensive metabolic panel of the mice after drug administration?

  1. What the prodrug effects on liver function? Mention the systemic half-life of the prodrugs? Also, reveal the aspects related to reticuloendothelial clearance of the drugs?

  2. What is the overall percent efficacy of prodrug in mice with the tumors?

  3. The manuscript can be revised further for grammatical and typological errors.

Author Response

  1. Did the authors assess any signs of distress and dehydration effects post prodrug(s) administration in mice?

Besides a red but harmless urine discoloration 1-2 days post administration, no other side-effects were observed upon doxorubicin, CBR-049 or CBR-050 therapy. Of note, CBR-049 dose and regimen was based on the maximum tolerated dose as determined by Casazza and colleagues. Above this dose of 154 mg/kg, toxicity (when observed) was represented by mild but reversible body weight loss (15%) and a temporary decrease of lymphocyte and neutrophil count.

The following sentences were added to the revised manuscript (results p.6): All treatments were well tolerated with mice showing stable body weight within ethically acceptable limits (Supplementary Figure S1.A). Besides a red but harmless urine discoloration 1-2 days post administration, no other side-effects were observed upon doxorubicin or prodrugs therapy in mice. Macroscopic evaluation of the hearts, liver and kidneys upon sacrification did not reveal any alteration nor signs of toxicity.

  1. What was the age of the mice at the time of PDX generation? Have authors evaluated the survival effects of prodrugs in mice beyond 28 days? If so, what probable physical changes were observed?

Mice were transplanted upon arrival at 7-8 weeks old and experiments started 3-4 weeks later, when majority of tumors reached ±100 mm3. For experiments with aldoxorubicin and CBR-049, part of mice were kept until 85 days but no effects on survival or physical changes could be observed for CBR-049-treated mice. As mentioned in the manuscript, several aldoxo-treated started showing hind limb paralysis weeks after aldoxo administration, which even led to sacrification in a few severe cases.  

  1. Did authors have collected the heart, liver, spleen, lung, brain, and kidney of mice after 28 days. Did you perform any histopathology analysis, to determine the effect of prodrugs on organ functions?

Heart, liver and kidneys were evaluated and collected upon sacrification of every mouse, but macroscopically no signs of toxicity could be observed upon necropsy. Histopathological analysis of these samples has not been performed. As mentioned in the discussion, more detailed toxicity studies were performed and recently published by Casazza and colleagues (DOI: 10.1158/1535-7163.MCT-21-0518).

  1. Did mice complete blood count analysis was performed during the course of the study? Had you looked into comprehensive metabolic panel of the mice after drug administration?

Unfortunately, we were not able to perform pharmacokinetic analysis on blood samples or tumor tissue. The following sentences were added (discussion p.12): Moreover, in this study no adverse events could be observed in mice treated with the experimental prodrugs, even though molar doses were 17-fold higher than those administered for doxo. Although the current study lacked more profound toxicological evaluation (i.e. blood analysis or histopathological evaluation of organs) and pharmacokinetic analysis, this has been covered in previous work [15].

  1. What the prodrug effects on liver function? Mention the systemic half-life of the prodrugs? Also, reveal the aspects related to reticuloendothelial clearance of the drugs?

See answer #4.

  1. What is the overall percent efficacy of prodrug in mice with the tumors?

CBR-049 showed efficacy compared to control in 10/10 (100%) of the xenografts investigated and compared to doxo in 9/10 (90%). CBR-050 showed efficacy compared to control in 1/6 (17%) of the xenografts investigated with this compound.

  1. The manuscript can be revised further for grammatical and typological errors.

The typos found have been corrected (e.g. in vivo -> in vivo)

Round 2

Reviewer 2 Report

Satisfied with the response.

Author Response

We would like to thank both reviewers for the time and effort that were dedicated to providing feedback on our manuscript. All comments were of great addition and significantly improved the manuscript.